# Radiomics in the Setting of Neoadjuvant Radiotherapy: A New Approach for Tailored Treatment

**DOI:** 10.3390/cancers13143590

**Published:** 2021-07-17

**Authors:** Valerio Nardone, Luca Boldrini, Roberta Grassi, Davide Franceschini, Ilaria Morelli, Carlotta Becherini, Mauro Loi, Daniela Greto, Isacco Desideri

**Affiliations:** 1Department of Precision Medicine, University of Campania “L. Vanvitelli”, 80138 Naples, Italy; v.nardone@hotmail.it (V.N.); grassi.roberta89@gmail.com (R.G.); 2Italian Society of Medical and Interventional Radiology (SIRM), SIRM Foundation, 20122 Milan, Italy; 3Radiation Oncology Unit, Fondazione Policlinico Universitario A. Gemelli IRCCS, 00168 Rome, Italy; luca.boldrini@policlinicogemelli.it; 4Radiotherapy and Radiosurgery Department, IRCCS Humanitas Research Hospital, via Manzoni 56, 20089 Milan, Italy; davide.franceschini@cancercenter.humanitas.it; 5Department of Biomedical, Experimental and Clinical Sciences “Mario Serio”, University of Florence, 50134 Florence, Italy; carlotta.becherini@libero.it; 6Radiation Oncology Unit, Azienda Ospedaliero Universitaria Careggi, 50139 Florence, Italy; mauro.loi82@gmail.com (M.L.); gretod@aou-careggi.toscana.it (D.G.); isacco.desideri@unifi.it (I.D.); 7Department of Experimental and Clinical Biomedical Sciences “Mario Serio”, University of Florence, 50134 Florence, Italy

**Keywords:** radiomics, neoadjuvant radiotherapy, texture analysis

## Abstract

**Simple Summary:**

This review based on a literature search aims at showing the impact of Texture Analysis in the prediction of response to neoadjuvant radiotherapy and/or chemoradiotherapy. The manuscript explores radiomics approaches in different fields of neoadjuvant radiotherapy, including esophageal cancer, lung cancer, sarcoma and rectal cancer in order to shed a light in the setting of neoadjuvant radiotherapy that can be used to tailor the best subsequent therapeutical strategy.

**Abstract:**

Introduction: Neoadjuvant radiotherapy is currently used mainly in locally advanced rectal cancer and sarcoma and in a subset of non-small cell lung cancer and esophageal cancer, whereas in other diseases it is under investigation. The evaluation of the efficacy of the induction strategy is made possible by performing imaging investigations before and after the neoadjuvant therapy and is usually challenging. In the last decade, texture analysis (TA) has been developed to help the radiologist to quantify and identify the parameters related to tumor heterogeneity, which cannot be appreciated by the naked eye. The aim of this narrative is to review the impact of TA on the prediction of response to neoadjuvant radiotherapy and or chemoradiotherapy. Materials and Methods: Key references were derived from a PubMed query. Hand searching and ClinicalTrials.gov were also used. Results: This paper contains a narrative report and a critical discussion of radiomics approaches in different fields of neoadjuvant radiotherapy, including esophageal cancer, lung cancer, sarcoma, and rectal cancer. Conclusions: Radiomics can shed a light on the setting of neoadjuvant therapies that can be used to tailor subsequent approaches or even to avoid surgery in the future. At the same, these results need to be validated in prospective and multicenter trials.

## 1. Introduction

Radiotherapy (RT) represents one of the most effective anticancer agents, which can be used either alone or in combination with other strategies (surgery, chemotherapy, and immunotherapy). It aims either to cure the patient (curative radiotherapy), to reduce the risk of locoregional relapse after surgery (adjuvant radiotherapy), to facilitate and to improve the results of surgery (neoadjuvant radiotherapy), or to relieve symptoms if a cure is not achievable (palliative radiotherapy). A neoadjuvant strategy, specifically, represents a type of induction therapy that is usually given as a first step to shrink a tumor before the main treatment, usually surgery, is given. In this setting, radiotherapy can be used either alone or in combination with chemotherapy.

This strategy is currently used mainly in locally advanced rectal cancer and sarcoma and in a subset of non-small cell lung cancer and esophageal cancer, whereas in other diseases it is still under investigation [1,2,3,4,5,6,7]. An evaluation of the efficacy of the induction strategy is made possible by performing imaging investigations before and after the neoadjuvant therapy. The imaging is evaluated by the radiologist and the response is usually classified following the RECIST guidelines [8,9].

In the last decade, texture analysis (TA) has been developed in order to help the radiologist to quantify and identify the parameters related to tumor heterogeneity that cannot be appreciated by the naked eye [10,11,12,13]. This analysis can use multiple mathematical models that are aimed to provide quantitative parameters within a selected image, which are called texture features. TA is performed employing a computer quantification of both the gray-level intensity and position of the pixels, and its use is being investigated in several fields [10,11,14,15,16,17,18,19,20,21,22,23,24,25,26,27]. More recently, a different approach to TA was developed, taking into consideration the variations in TA parameters at different acquisition times. This approach is usually called delta texture analysis or delta radiomics (D-TA) [28,29,30,31]. With this method, it is possible to investigate the role of TA variations after therapy (usually neoadjuvant chemotherapy or radiotherapy) or shortly after the beginning of the therapy.

Herein, we will discuss the impact of TA in the prediction of response to neoadjuvant radiotherapy and or chemoradiotherapy, focusing on rectal cancer, sarcoma, lung cancer, and esophageal cancer. Following a literature search, we will provide a narrative overview of these topics.

## 2. Materials and Methods

### 2.1. Evidence Acquisition

An electronic literature search was conducted in the PubMed database for English articles published up to 30 April 2021. Boolean operators (OR, AND) were used to combine the following search terms: “texture analysis”, “radiomics”, “neoadjuvant”, “neoadjuvant radiotherapy”, “induction radiotherapy”, “rectal cancer”, “sarcoma”, “esophageal cancer”, and “lung cancer”. Two independent reviewers (V.N. and L.B.) screened the titles and abstracts and performed the final article selection. Any discrepancy was resolved by discussion with a third reviewer (I.D.). Meeting proceedings (European Society of Medical Oncology—ESMO; European SocieTy for Radiotherapy and Oncology—ESTRO; American Society of Clinical Oncology—ASCO; and American Society for Radiation Oncology—ASTRO), trial registries (ClinicalTrials.gov), reference lists of published studies, review articles, and relevant books were also considered.

### 2.2. Texture Analysis: Workflow and Definition

Radiomics aims at the extraction of quantitative parameters from medical imaging. Different imaging techniques can be used for this purpose (CT, MRI, PET/CT, and US) and the features extracted can be also correlated with other cancer characteristics, such as tumor markers, metabolic activity of the tumor volume, tumor microenvironment, and tumor infiltrating lymphocytes. The integration of radiomic features and known clinical parameters with approaches such as radiogenomics is particularly important to better understand the clinical meaning of texture parameters [10,11,12,13].

Radiomics can be applied to both solid and non-solid tumors, although in the latter the gross tumor volume cannot be contoured and the texture analysis would include other endpoints (such as chemotherapy or radiotherapy toxicity or organ invasion diagnosis) [32,33]. 

The texture features can be predefined (feature-based radiomics) or identified and generated by computational models (deep learning-based radiomics) [10,11,14,15,16,17,18,19,20,21,22,23,24,25,26,27]. A feature-based radiomics workflow usually relies on image preprocessing and region-of-interest (ROI) segmentation, followed by feature extraction. Radiomics features are based on a huge number of different mathematical operations; thus, a large number (even more than 1000) of features can be extracted from a single ROI. These features can be divided in different subgroups, as follows:
(a)*Shape features*: these features refer to the geometric properties of the ROI (volume, diameter, sphericity, and compacity);(b)*Histogram-based features:* these features are calculated from the general histogram of the Hounsfield Unit (HU) of the ROI, such as the mean, median, skewness, and kurtosis. These features do not consider the spatial orientation of the voxels;(c)*Second-order texture features:* these features consider the statistical relationship between neighboring voxels or groups of voxels within the segmented lesion. These features can be extracted from several matrices, such as the gray-level co-occurrence matrix (GLCM), gray-level run-length matrix (GRLLM), and neighborhood gray-level different matrix (NGLDM);(d)*Higher-order texture features:* these parameters use additional image filters, using specific mathematical transformations that can highlight specific aspects of the ROI. The filter used can be wavelet or Fourier transforms, fractal analysis, and Laplacian of Gaussian;

Deep learning–based radiomics, conversely, use neural networks or auto-encoders to generate and identify important features from input data [34]. Deep learning is an application of Artificial intelligence (AI) that represents a set of computational algorithms that are able to learn the patterns in the data provided in order to make predictions on other data sets. The application areas include image segmentation and phenotyping, radiomic signature discovery, clinical outcome prediction, and many other processes. With this AI approach, radiomics can handle a massive amount of data and at the same time can directly analyze the imaging to design the proper radiomic features.

## 3. Results

### 3.1. Texture Analysis in Neoadjuvant Radiotherapy—A Focus on Esophageal Cancer

A trimodal approach with neoadjuvant chemoradiotherapy (nCRT) followed by surgery is the treatment of choice in locally advanced esophageal cancer (EC).

A clinical evaluation of the response is made with different imaging techniques (MRI, CT, and PET) and can classify the response according to the RECIST criteria [35,36,37,38,39,40]. With these premises, radiomic analysis has been used in this setting with the aim to predict the response of the neoadjuvant strategy [37].

In patients treated for locally advanced EC, temporal changes in tumor volume (before and after nCRT) were correlated with pathological response, although the predictive value of this parameter was modest, with no correlation with overall survival [35].

Sun et al. [37] found that several histogram parameters calculated on DCE-MRI can be used in evaluating and predicting the nCRT response.

CT images are mostly used to extract morphologic information of EC, but recent studies suggest that quantitative image features can provide additional information correlated to tumor response and prognosis [41,42,43].

On the other hand, metabolic imaging has been found to be more accurate in evaluating the response to nCRT. Most 18F-FDG PET studies in EC quantify metabolic tumor activity solely by using the maximum standardized uptake value (SUVmax) [44,45]. However, this approach does not characterize the total activity nor heterogeneity of the 18F-FDG uptake for the entire tumor [46,47]. Recent studies suggest that spatial image information provides additional information than SUVmax [48,49,50,51].

It has been hypothesized that tumors could be rendered more homogeneous following treatment due to a reduction in cellular density and interstitial pressure and to normalization of the vasculature with improved intra-tumor perfusion and oxygenation [43]. Studies that performed imaging before and after treatment reported that tumor heterogeneity generally decreased following treatment [43,52]. Simoni et al. found that metabolic TV (MTV) in poor responders was significantly higher than in good responders (38.6 mL vs. 17.7 mL, *p* = 0.02) [53].

Yip et al. reported that delta-radiomics calculated within MTV60% were the least correlated with pathological response. Instead, temporal changes in tumor 18F-FDG distribution after nCRT assessed by delta-second-order radiomics (ΔRLM textures) were correlated to the OS [54].

In a study of Beukinga et al., the most predictive textural features were Long Run Low Gray-Level (LRLGLe)-PET and Run-Percentage (RP)-CT. LRLGLe-PET depends on long runs (coarse texture) with low gray levels and was higher (i.e., low and homogeneous 18F-FDG uptake) for complete responders and lower (i.e., high and heterogeneous 18F-FDG uptake) for incomplete responders, possibly due to tumor hypoxia and necrosis [55].

In a French study, Hatt et al. assessed the value of MTV, entropy, dissimilarity, high-intensity large-area emphasis, and zone percentage for the prediction of OS of 112 patients with EC who underwent definitive CRT or nCRT followed by surgery [56]. MTV and heterogeneity had a less prognostic value in EC (vs. non-small cell lung cancer), which was attributed to smaller overall volumes. The local dissimilarity parameter appeared most predictive for OS.

Current research is focused on a correlation between the biological tumor markers from pre-treatment tumor biopsies and F-FDG PET-based radiomic features. The prediction model from Beukinga et al. tried to correlate HER2 and the CD44 expression and F-FDG PET-based radiomic features in predicting a complete response (pCR) to nCRT; it showed promising ability to identify pCR [57]. Analysis of texture within tumors on medical imaging, such as CT, MRI, and PET, is emerging as a potential biomarker to predict prognosis and treatment response in patients with EC. These findings could increase our capability to predict and evaluate the nCRT response and optimize treatment planning.

### 3.2. Texture Analysis in Neoadjuvant Radiotherapy—A Focus on Lung Cancer

Lung cancer represents the leading cause of cancer death and its treatment rely on different strategies, often used in combination, such as radiotherapy, surgery, chemotherapy, and immunotherapy [58,59,60,61]. Medical imaging is pivotal in all the different phases of lung cancer management, from diagnosis to the evaluation of the efficacy of different approaches, as well as in the subsequent follow-up [62,63,64,65,66,67,68,69].

A significant body of literature is available regarding the role of textural analysis in lung cancer. Radiomic features have been used both for diagnosis (discrimination of malignant nature of lung nodules, histology prediction, and genomic analysis) and for outcome prediction, particularly after definitive chemo-radiotherapy (CT-RT) or stereotactic body radiation therapy (SBRT) [70,71,72,73,74,75,76,77,78,79]. On the other hand, few studies focused on neoadjuvant radiotherapy, possibly because of the controversial benefit of adding surgery after CT-RT [80,81]. The largest available series included 127 stage II–III NSCLC patients treated with neoadjuvant chemoradiotherapy (CT-RT) followed by surgical resection. Texture analysis was conducted on primary tumors with features extracted from planning CT. The endpoint of the analysis was a pathologic response and gross residual disease. Seven features (one shape feature, three second-order features, and three higher-order features) were predictive for pathologic gross residual disease (AUC > 0.6, *p*-value < 0.05) and one for pathologic complete response (one higher-order feature) (AUC = 0.63, *p*-value = 0.01). These features suggested that a spherical disproportionality of the primary tumor site (i.e., more complex shape) was a predictor of pathological response, while spherical tumors and tumors with large flat zones were more likely to be linked with residual disease [82]. The same group conducted another analysis on a similar population, analyzing also features from pathologic lymph nodes. Radiomics analysis was performed on 85 primary tumors and 178 lymph nodes with the same endpoints as above (pathologic response and gross residual disease). A mix of clinical variables and conventional and radiomic features were used to create a predictive model. The model built from the radiomic features was the best at predicting pathologic response (AUC 0.68, *p* < 0.05), while a mixed clinical and radiomic model was better performing for gross residual disease (AUC 0.73, *p* < 0.05). The authors concluded that the features extracted from the lymph nodes were more informative than those derived by primary tumors [83].

A small study analyzing PET-CT included 13 patients on a prospective clinical trial of trimodal therapy for resectable locally advanced NSCLC. A true texture analysis was not performed; however, pre- and post-chemo-radiotherapy SUV was not correlated with any clinical endpoint (pathologic complete response, PFS, and OS) [84].

Chong et al. performed a texture analysis on a mixed population with NSCLC treated with neoadjuvant chemo-radiotherapy (CT-RT) or neoadjuvant EGFR-inhibitor. Focusing on the CT-RT group, 28 patients were included in the study. A pre-operative CT was used for analysis, and the endpoint again was a pathologic response. In the univariate analysis, tumor volume, mass, kurtosis, and skewness were significant predictors of pathologic response, although only kurtosis maintained its significance in the multivariate analysis (OR 1.107, *p* = 0.009). ROC analysis showed that the AUC for kurtosis was 0.943 and that the optimal cut-off value of percent change of kurtosis for predicting pathologic response was less than −23 (sensitivity, 87.5%; specificity, 84.3%) [85].

More recently, Khorrami et al. conducted a textural analysis on 90 stage IIIA NSCLC patients treated with neoadjuvant CT-RT, followed by surgical resection. Pretreatment CT scans were used for features extraction. Patients were randomly split into two sets, one for training and one for testing.

Interestingly, the authors analyzed the features not only from primary tumors, but also from the peritumoral region (defined as a 15 mm dilation from primary nodule) to identify a possible predicting role of the tumor microenvironment.

Thirteen intratumoral and peritumoral radiomic texture features were found to be predictive of major pathologic response and were used to define a classifier with an AUC of 0.90 ± 0.025 within the training set, and an AUC = 0.86 in the test set. This signature was also predictive for OS (HR = 11.18, 95% CI = 3.17, 44.1; *p*-value = 0.008) and DFS (HR = 2.78, 95% CI = 1.11, 4.12; *p*-value = 0.0042) in the testing set. The combination of peritumoral and intratumoral features performed better than a clustering built only on intratumoral features.

The results of this study highlight that heterogeneous enhancement and disruption of textural patterns within and outside the nodules (better identified by Law Laplacian and Laws features) can predict not only response, but also patients’ prognosis [86].

Identifying patients who respond completely to chemoradiation and who do not require additional invasive local therapy, also considering the controversial role of surgery in this scenario, is an unmet clinical need.

The intrinsic potential of texture analysis in predicting a response after neoadjuvant CT-RT is plain. However, available studies are not enough to consider radiomics as a ready and easily accessible clinical biomarker. Larger studies, with a higher number of patients and with external validation, are needed before radiomic analysis could be clinically implemented. Analyzing available data, a special attention in future research should be paid to nodal and peritumoral features, which are potentially even more predictive than primary tumor analysis.

### 3.3. Texture Analysis in Neoadjuvant Radiotherapy—A Focus on Sarcoma

Soft tissue sarcomas (STS) are a group of mesenchymal malignancies encompassing a wide array of distinct clinical entities, including over 50 different histological subtypes [87]. Due to their rarity and non-specific clinical presentation (mostly consisting of slowly-growing indolent swellings in limbs or trunk), a differential diagnosis with benign tumors is challenging and may result in a significant delay to curative treatment, consisting of wide surgical excision in absence of metastatic dissemination [87,88]. Furthermore, their intrinsic heterogeneity is also displayed by variable levels of genomic profile complexity and structural architecture, resulting in different clinical presentations and inconsistent responses to standard treatments [88,89,90]. Unfortunately, one among the major predictors of outcome in operable patients—tumor histological grading—may be underestimated by tumor biopsy [91]. This is a relevant issue, since tumor grade determination may guide treatment management: low-grade STSs may be treated with planned marginal excision, sparing patients from more extensive surgery [92], while detection of high-grade disease may serve as a decision criteria for pre-operative treatment intensification [93,94]. Moreover, while the additional benefit of adjuvant chemotherapy is questioned and limited to a subset of high-risk patients, no consensual criteria have been provided to define this population; clinical [95] and genomic [96] classifiers have been proposed, although their use have not been implemented in clinical practice yet.

Hence, there is an urgent need to identify novel and readily available biomarkers to improve diagnosis, predict the disease course, and possibly define a tailored pattern of care in patients at high risk of disease relapse [88]. Radiomics may provide a valuable source of information in STS patients. Due to its non-invasive nature, radiomics may quickly discern benign from malignant lesions, timely addressing the need for further investigation; secondarily, it may provide an appropriate histopathologic grade determination on the whole tumor bulk, thus overcoming the limitations of tumor biopsy in accounting for intratumor heterogeneity; finally, identification of specific signatures correlated with outcome may allow prediction of relapse risk and eligibility for further interventions.

Improvement in differential diagnosis of malignancy have been reported by several authors [97,98]. In particular radiomics-based differentiation between soft-tissue lipoma and well-differentiated liposarcoma [99,100] was demonstrated despite similar radiologic and pathologic presentation, often requiring molecular analysis of MDM2 amplification status; interestingly, superior performance of a machine-learning classifier as compared to trained radiologists has been shown [101]. Similarly, radiomic features allowed to distinguish myxoma from myxofibrosarcoma [102] and atypical leiomyoma from uterine sarcoma [103,104,105].

Machine-learning algorithms proved also useful in the prediction of histopathologic grade. Using CT- or MRI-based radiomics, different signatures associated with high-grade disease were identified [106,107,108,109,110,111,112]; in some cases, integration with clinical features resulted in the establishment of a prognostic nomogram for risk stratification [108].

A radiomic approach may be used to improve prediction of patients’ outcome. RM texture analysis alone [113,114] or combined with PET/CT metabolic data [115,116] was associated with metastatic relapse and specific signatures were identified for prediction of survival [117,118]. Radiomic analysis was also applied on surveillance MRI in patients undergoing follow-up after surgical resection [119], resulting in improved detection and characterization of local recurrence [120].

Concerning prediction of sensitivity to treatment, preliminary results reported adequate concordance between radiomic features and response to neoadjuvant chemotherapy [121] or chemoradiation [122], as well as exclusive chemotherapy in unresectable patients [119,123].

On the other hand, while radiomics analysis correlated with adipocytic maturation following neoadjuvant chemotherapy in myxoid/round cells liposarcomas, it occurred independently from the chemotherapy regimen and was not correlated with metastasis-free survival [124]; similarly, radiomics was insufficient to predict the response from hypofractionated pre-operative RT [125].

Despite promising preliminary results, no radiomic signature for sarcoma has been implemented to date in clinical practice. Multiple issues have been detected in the pathway to translation from exploratory analysis into standard of care, ranging from lack of external validation and replicability, consistency, and cost-effectiveness of the imaging biomarkers proposed at present [126]. While these shortcomings are partially motivated by the complex workflow and relevant technical requirements needed for the development and elaboration of radiomic signatures, further effort is required for successful translation into clinical use.

### 3.4. Texture Analysis in Neoadjuvant Radiotherapy—A Focus on Rectal Cancer

Adequately predicting the response to neoadjuvant chemoradiotherapy (nCRT) in rectal cancer represents a key factor in order to achieve a fully personalized treatment approach, especially in the case of locally advanced disease presentation, which constitute a well-established indication for radiation therapy [127]. The identification of biomarkers able to predict patients’ outcomes (i.e., response to nCRT) currently represents a priority in the radiotherapy research landscape, and imaging-derived response predictors are a topic of active investigation [128,129,130,131,132,133].

Thanks to the extensive use of MRI as a staging imaging technique for this specific disease, the imaging community rapidly focused its research efforts on it and the first preliminary observations have been published since the early 2000′s. These first studies generally aimed to correlate the response to neoadjuvant treatments with the Gross Tumor Volume’s (GTV) morphological modifications, mainly taking advantage of the high image quality provided by MRI [134,135,136] and also proposing to address the metabolic information disclosed by PET-CT staging imaging [137,138].

The recent development of TA and radiomics-based image analysis have introduced new predictive modelling techniques and offered promising insights for quantitative imaging characterization of rectal cancer, when compared to the mere volumetric assessment [139].

Most of the papers available in the literature focus on the response to nCRT prediction, using several imaging and modelling techniques. Handcrafted radiomics is used in the largest majority of these studies, achieving overall satisfactory results even when compared to more advanced machine learning approaches (i.e., deep learning) [34].

Nevertheless, the obtained level of evidence is still unfortunately low and inconclusive, mainly due to the overall low quality of these studies, as pointed out by two recent reviews that highlight the limited replicability and reproducibility of the available models [140,141].

Furthermore, the largest majority of the published papers does not meet the minimal technical quality criteria, such as the ones proposed by the RQS and QUADAS-2 standards [142,143]. In this context, it has also to be said that this issue suffers from the initial enthusiasm that researchers have addressed to radiomics and TA analysis, publishing several exploratory papers on very limited cohorts of patients and often using non-standardized, and thus not reproducible analysis pipelines.

Furthermore, only few models are based on a properly selected reduced number of features that make their applicability easier and more straightforward, while the majority gathers hundreds of variables.

Among the different imaging techniques used, it is evident that staging MRI represents the one of choice for TA and radiomics studies in general, primarily thanks to the large availability of images and to the prominent current role fulfilled by MRI in the framework of the international rectal cancer management guidelines as the imaging gold standard for disease staging [144].

Response to therapy (e.g., tumor downstaging and pathological complete response, pCR) represents the most common outcomes of these models, whose main part focuses on the use of T2w and Diffusion Weighted Imaging (DWI), while the use of Dynamic Contrast Enhanced (DCE) images is still less common [141].

Interestingly, the largest majority of the published MRI radiomics studies takes into account histogram features (considered alone or in more advanced models based also on textural, shape, and filtered ones), supporting systematic investigations in this direction [28,145,146,147,148,149,150,151,152,153,154,155,156,157,158,159,160,161,162,163,164,165,166,167,168,169,170,171,172].

Unfortunately, no consensus has been reached in the literature about the role and the potential biological correlates of the radiomics features that have been identified in the studies published so far. In their recent review, Staal and colleagues systematically point out this pitfall, shedding a light on the complex translational impact of the published observations, but also pointing out interesting correlations that deserve to be further explored, such as the one between tumor heterogeneity, image homogeneity, and generally favorable patients’ outcomes [141].

Besides traditional diagnostic imaging, the recent introduction of MR-guided radiotherapy (MRgRT) paves the way to innovative clinical approaches, including promising modelling applications for hybrid imaging TA and radiomics analysis [173,174,175]. Although the published evidence is still scarce and exploratory, rectal cancer hybrid imaging appears indeed to have significant potentialities for the set-up of radiomics-based prediction models. A first hypothesis generating experience identified two delta radiomics features (ΔL_least and Δglnu) as possible candidate predictors of a complete response (cCR) after rectal cancer neoadjuvant therapy using 0.35 T setup images [28]. The model was then externally validated using independent cohorts and achieved remarkable performances for both cCR and pCR (accuracy for cCR prediction: ΔLleast = 81% and Δglnu = 63%; accuracy for pCR prediction: ΔLleast 79% and Δglnu = 40%), confirming the interest in applying radiomics to hybrid MR images [150]. These observations may lead to a new generation of innovative trials, personalizing a patient’s treatment based on imaging-based prediction results (clinical trials NCT04815694).

Besides MRI-based investigations, a more limited number of papers explore the performances of PET-CT for response prediction, obtaining overall unsatisfactory results in terms of prediction performances, probably due to technical reasons relative to PET imaging formats that still remain unsolved [176,177,178,179,180].

Radiomics and TA therefore represent a promising field of investigation for rectal cancer and may support the continuous development of accurate prediction models, integrating the variables extracted from different omics domains in a holistic and fully personalized treatment approach [181,182].

## 4. Future Directions and Conclusions

All the above-mentioned studies have shown the promising direction of using radiomics in the field of neoadjuvant therapies. Despite an impressive number of retrospective studies, the number of prospective clinical trials investigating the potential role of texture analysis in neoadjuvant radiotherapy remains inexplicably low (see Table 1).

There still are major limitations to solve before TA can be successfully applied in the clinical management of cancer patients. More specifically, the current major pitfalls in TA are the lack of image standardization, the lack of feature extraction standardization, the problems related to data sharing, and the distrust of the clinicians in the black box approach.

All the variations in scanning devices, acquisition protocols, and parameters of reconstructions may impact on the process of feature extraction and analysis. Although different standardization techniques may calibrate and overcome partially this problem [183,184,185,186], the ideal solution is to standardize the image acquisition in dedicated prospective trials.

Similarly, the process of feature extraction is often challenging and can lead to several biases that impact the reliability of the TA parameters among different institutions. Hopefully, with the help of artificial intelligence and the optimization of machine learning approaches, the limitations of feature extraction (ROI segmentation, feature selection, and endpoint correlation) should be overcome in the foreseeable future [187,188,189,190,191].

At the same time, sample size in TA research represents a drawback in current research, as most studies cited above cover a limited number of patients. The use of big data, instead, could overcome this problem, and could ensure a higher statistical significance in the interpretation of the data. Thus, several efforts should be made in the development of high-number and high-quality shared databases in the future. These datasets require joint efforts by both companies and institutions, such as the Cancer Learning Intelligence Network for Quality and Flatiron Health and the Cancer Imaging Archive. At the same time, the development and maintenance of such shared databases require high standards of security, in order to respect patients’ privacy and actual law order in terms of data sharing and data privacy.

The texture parameters are somehow difficult to describe and to refer to known clinical variables; as a consequence, TA looks like a black box to clinicians, as the connection between the TA parameters and the endpoints is not fully understood. Nonetheless, more should be done in order to correlate radiomics to the underlying clinical and molecular connotation with approaches such as radiogenomics; at the same time, more should be learned about artificial intelligence and radiomics by the clinicians [192,193].

Currently, many researchers are investigating methods to diminish the black box perception [194]. These two processes together can help to empty the gap and to remove the black box uncertainties, in order to promote TA towards clinical practice.

Despite these pitfalls, TA analysis can still aspire to become an optimal surrogate biomarker. In this context, research efforts should be concentrated precisely on the field of neoadjuvant therapies. In this setting, in fact, it is easier to test different approaches because the endpoint is usually the pathological response, which is more immediate to accomplish. In this field, the clinicians could trust more easily in radiomics, if they are called to test an effective and reliable model. For this reason, neoadjuvant therapy represents the ideal playground for TA and a higher number of prospective trials should be ideated and conducted in the next future.

Finally, future combination of radiomics analysis in the setting of immunotherapy is needed in the next years. This interesting challenge is particularly important, as imaging is always used in the clinical management of cancer patients, and TA may in the future provide useful information regarding the molecular characteristics of cancer, which may differ from a biopsy due to the accumulation of multiple genetic mutations and epigenetic alterations.

## Figures and Tables

**Table 1 cancers-13-03590-t001:** Prospective clinical trials investigating the potential role of texture analysis in neoadjuvant radiotherapy.

NCT Number	Study Type	Cancer Type	Trial Design	Contacts and Locations	Trial Design
NCT02439086	Interventional	Rectal Cancer	18F-FDG-PET-CT and texture analysis of MRI performed 9 weeks after Neoadjuvant Chemo-radiotherapy in patients with locally advanced rectal cancer to test the ability to identify patients with Complete Response.	Medhat S Alaker, Colchester General Hospital, UK	Single arm, patients will have 2 PET CT scans: one before radiotherapy, and one 9 weeks after.
NCT04273477	Observational	Rectal cancer	Radiomics prediction model to predict the tumor response to neoadjuvant chemoradiotherapy (nCRT) before the nCRT is administered.	Xiangbo Wan, Sun Yat-sen University, China	Multicenter, prospective, observational clinical study, evaluating MRI performed before nCRT.
NCT03238885	Observational	Rectal cancer	Develop and validate a radiomics model for individualized pCR evaluation after CRT in patients. The ultimate aim is to select appropriate LARC patients for omission of surgery.	Sun Ying-Shi, Beijing Cancer Hospital, China	Prospective, observational cohort study, investigating 3T MRI performed before and after nCRT.
NCT04489368	Observational	Esophagus Cancer	Develop models to predict pCR based on pre-neoadjuvant imaging modalities	Kundan S Chufal, Rajiv Gandhi Cancer Institute & Research Center, India	Prospective, observational cohort study, investigating imaging performed before nCRT.
NCT04278274	Observational	Rectal Cancer	Evaluation of Post-Neoadjuvant Treatment MRI Based AI System to Predict Pathologic Complete Response for Patients With Rectal Cancer.	Xiangbo Wan, Sun Yat-sen University, China	Multicenter, prospective, observational clinical study, evaluating MRI performed after nCRT and before surgery.
NCT04815694	Interventional	Rectal Cancer	Investigate the impact of dose escalation in rectal cancer, identifying the poor responder cases using the early tumor regression index during the course of radiotherapy and increasing the prescribed dose in these patients. Secondary endpoint is prospective validation of delta radiomics MR-guide Radiotherapy model	Giuditta Chiloiro, Fondazione Policlinico Universitario A.Gemelli IRCCS, Italy	Interventional, two arms. In experimental arm RT Dose escalation will be performed in patients based on Early Regression Index values calculated at second week on nCRT
NCT04359732	Interventional	Esophagus Cancer	Prediction of Assessment of Response to Neoadjuvant Chemo-Radio-Therapy (nCRT) for Esophageal and Gastroesophageal Junction Cancer (GEJ) Using a Fully Integrated PET/MRI	Francesco De Cobelli, IRCCS San Raffaele, Italy	Interventional, single arm. An additional intermediate 18-FDG PET/MRI will be performed during nCRT.
NCT03237130	Observational	Esophagus Cancer	Establishment of an image feature extraction and selection method for identifying lymph node metastasis of esophageal cancer	Ying-Shi Sun, Dept.Radiology, Peking University Cancer Hospital, China	Prospective, observational cohort study. Each patient will receive preoperative enhanced chest CT examination, and their CT images will be used for analysis
NCT04090450	Observational	Rectal Cancer	Retrospective study using images acquired routinely for diagnosis of rectal cancer to see if these could be used to predict responses to radiotherapy treatment and if it can, whether the treatment can be optimized to produce better outcome for patients.	Peter Mbanu, University of Manchester, UK	Retrospective, observational cohort study and will recruit patients who have had nCRT for rectal cancer. MR radiomics features will be analyzed.
NCT03029793	Observational	Esophagus Cancer	Determine whether combination of molecular and biomarkers with functional imaging can predict pathologic response and clinical outcomes in squamous esophageal cancer patients who undergo trimodal therapy which includes neoadjuvant chemoradiotherapy and surgery	Li-Na Zhao, Air Force Military Medical University, China	Prospective, observational cohort study, investigating imaging performed before nCRT. The study will collect both tissue samples and imaging in locally advanced esophageal cancer patients.
NCT04207918	Interventional	Esophagus Cancer	Evaluate the 1-year local tumor control rates after the targeted therapy of intensity-modulated radiation therapy synchronized chemotherapy with nimotuzumab. Secondary texture analysis of CT and MRI simulation imaging in predicting tumor response rate is included.	Wang Xin, Chinese Academy of Medical Sciences, China	Experimental, single arm, nCRT arm receives RT concurrently with S-1 and Nimotuzumab. Texture analysis of CT and/or MRI simulation will be analyzed in predicting tumor response rate and prognosis.

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
