# Peer review of "Radiomics in the Setting of Neoadjuvant Radiotherapy: A New Approach for Tailored Treatment"

_cancers, 2021, doi:10.3390/cancers13143590_

Round 1
Reviewer 1 Report
The Authors provide a comprehensive review of the utility and application of radiomics in the field of neoadjuvant therapies. The paper is extensive with a large body of literature. However, I have the following concerns.
General comments:
- the introduction should be streamlined with a more fluent English
- the authors have focused on radiomics in the neoadjuvant radiotherapy setting. They left out breast cancer for which neoadjuvant chemotherapy is consolidated. Moreover, neoadjuvant therapies have an important role also in gastric and pancreatic cancer. I would recommend that the authors also include these chapters.
- in my opinion the authors should dedicate a subchapter in the methods section to describe in details what texture analysis is. They should describe at a minimum basic technical principles of texture analysis. They should help the reader to understand what exactly texture analysis is. Radiomics features vary a lot, from simple volume analysis to complex mathematical derived parameters, such as ktrans, skewness, kurtosis etc. The reader should be briefly introduced to the differences among these parameters.
- The chapter on esophageal cancer is a little bit confused and difficult to read. The authors introduce MR, then CT and PET. They present studies about tumor volume (TV) in different paragraphs. This section should be reorganized and simplified.
Specific comments:
- Abstract: The incipit focuses on neoadjuvant radiotherapy while the title is generic about neoadjuvant therapy. Consider adding radiotherapy to the title, otherwise other important chapters such as breast, gastric and pancreatic cancer should be treated.
- Introduction: again, the authors focus on neoadjuvant radiotherapy. It should be stressed the role of chemotherapy in other type of cancers (breast, gastric). Alternatively, the focus on radiotherapy should be included in the title
- Materials and Methods: why was “neoadjuvant chemotherapy” not included in the search? I would suggest to add a subchapter introducing radiomic principles and main features. Who are reviewers BOH?
- Results: Esophageal cancer: too much detailed information is given without adequate explanation and the text is difficult to read. The authors should provide a clear explanation of the findings presented.
Lung cancer, sarcoma and rectal cancer: ok
- Conclusions: “Despite an impressive number of retrospective studies, the number of prospective clinical trials investigating the potential role of texture analysis in neoadjuvant radiotherapy remains inexplicably low” AND “In this regard, despite an impressive number of retrospective published papers, the number of prospective trials investigating the role of TA in neoadjuvant cancer therapies is inexplicably low”: Please remove one of the two repetitive sentences.
- Table 1: not received!!!, please add to the text
Author Response
The Authors provide a comprehensive review of the utility and application of radiomics in the field of neoadjuvant therapies. The paper is extensive with a large body of literature. However, I have the following concerns.
- We thank the Reviewer for his/her efforts reading end evaluating our manuscript.
General comments:
- the introduction should be streamlined with a more fluent English
- We thank the Reviewer for his/her time spent reading and evaluating our manuscript. We believe that thank to his/her suggestions our manuscript has improved.
We have rewritten the Introduction section.
- the authors have focused on radiomics in the neoadjuvant radiotherapy setting. They left out breast cancer for which neoadjuvant chemotherapy is consolidated. Moreover, neoadjuvant therapies have an important role also in gastric and pancreatic cancer. I would recommend that the authors also include these chapters.
- We have specified the focusing on neoadjuvant radiotherapy in the title, following the Reviewer suggestion.
- in my opinion the authors should dedicate a subchapter in the methods section to describe in details what texture analysis is. They should describe at a minimum basic technical principles of texture analysis. They should help the reader to understand what exactly texture analysis is. Radiomics features vary a lot, from simple volume analysis to complex mathematical derived parameters, such as ktrans, skewness, kurtosis etc. The reader should be briefly introduced to the differences among these parameters.
- We have added a subchapter in the Methods section describing the basic principles of texture analysis.
- The chapter on esophageal cancer is a little bit confused and difficult to read. The authors introduce MR, then CT and PET. They present studies about tumor volume (TV) in different paragraphs. This section should be reorganized and simplified.
- We have rewritten the paragraph on esophageal cancer, following the Reviewer’ suggestions.
Specific comments:
- Abstract: The incipit focuses on neoadjuvant radiotherapy while the title is generic about neoadjuvant therapy. Consider adding radiotherapy to the title, otherwise other important chapters such as breast, gastric and pancreatic cancer should be treated.
A- We have specified in the title the focus of the manuscript;
- Introduction: again, the authors focus on neoadjuvant radiotherapy. It should be stressed the role of chemotherapy in other type of cancers (breast, gastric). Alternatively, the focus on radiotherapy should be included in the title
A- We have specified in the title the focus of the manuscript;
- Materials and Methods: why was “neoadjuvant chemotherapy” not included in the search? I would suggest to add a subchapter introducing radiomic principles and main features. Who are reviewers BOH?
A- We have added the requested subchapeter explaining the basic principles of texture analysis. We have specied the acronyms of the reviewers.
- Results: Esophageal cancer: too much detailed information is given without adequate explanation and the text is difficult to read. The authors should provide a clear explanation of the findings presented.
A- We have rewritten the paragraph of Esophageal cancer.
Lung cancer, sarcoma and rectal cancer: ok
- Conclusions: “Despite an impressive number of retrospective studies, the number of prospective clinical trials investigating the potential role of texture analysis in neoadjuvant radiotherapy remains inexplicably low” AND “In this regard, despite an impressive number of retrospective published papers, the number of prospective trials investigating the role of TA in neoadjuvant cancer therapies is inexplicably low”: Please remove one of the two repetitive sentences.
A- We have removed the latter sentence.
- Table 1: not received!!!, please add to the text
A- We have added the Table to the manuscript.
Reviewer 2 Report
In this review, the authors discuss the impact of "texture analysis" in the prediction of response to neoadjuvant radiotherapy and / or chemoradiotherapy, focusing on rectal cancer, sarcoma, lung cancer and esophageal cancer.
The review is well written, and the search correctly performed.
Few comments:
- An introductive paragraph describing how machine-learning supports "radiomics", and how radiomics integrate both the imaging and therapy field in tumor diagnosis and tumor management, would be of great utility for any sort of reader, with high or low background in this field.
- In my opinion the authors could shortly describe how the different imaging metodologies can support texture analysis and which parameters (physiological, metabolic, tumor microenvironment) have to be taken into account when performing such analysis, and how they can be integrated to provide a reliable description of the efficacy of a certain therapy.
- Does a texture analysis apply only to solid tumors? Please elucidate this point.
- In the paragraph entitled "Texture analysis in neoadjuvant radiotherapy – focus on Lung cancer ", the authors mention that ".....Out of fifteen radiomic features selected....". It would be better to describe which radiomic features are taken into consideration when performing such analysis and put them in relation to the pathophysiology characteristics of the diseases. A table would be explicative.
- Immunotherapy represents nowadays a hot area of intervention in NSCLC therapy and not only, which relies on the ability of modulating immune response via immune check-point inhibitors for example. The authors should discuss more in detail the challenge of applying "radiomics" in the context of this kind of therapy approach.
Author Response
In this review, the authors discuss the impact of "texture analysis" in the prediction of response to neoadjuvant radiotherapy and / or chemoradiotherapy, focusing on rectal cancer, sarcoma, lung cancer and esophageal cancer.
The review is well written, and the search correctly performed.
- We thank the Reviewer for his/her efforts reading end evaluating our manuscript.
Few comments:
- An introductive paragraph describing how machine-learning supports "radiomics", and how radiomics integrate both the imaging and therapy field in tumor diagnosis and tumor management, would be of great utility for any sort of reader, with high or low background in this field.
A- We have written a subchapter in the Methods section describing the basic principle of radiomics.
- In my opinion the authors could shortly describe how the different imaging metodologies can support texture analysis and which parameters (physiological, metabolic, tumor microenvironment) have to be taken into account when performing such analysis, and how they can be integrated to provide a reliable description of the efficacy of a certain therapy.
A- We have written a subchapter in the Methods section describing the basic principle of radiomics.
- Does a texture analysis apply only to solid tumors? Please elucidate this point.
A- Radiomics can be applied to both solid tumors and non solid tumors, although in the latter the gross tumor volume cannot be contoured and the texture analysis would include other endpoints (such as chemotherapy or radiotherapy toxicity, or organ invasion diagnosis). We have specified this concept in the manuscript.
- In the paragraph entitled "Texture analysis in neoadjuvant radiotherapy – focus on Lung cancer ", the authors mention that ".....Out of fifteen radiomic features selected....". It would be better to describe which radiomic features are taken into consideration when performing such analysis and put them in relation to the pathophysiology characteristics of the diseases. A table would be explicative.
A- We have rewritten the description of this paragraph.
- Immunotherapy represents nowadays a hot area of intervention in NSCLC therapy and not only, which relies on the ability of modulating immune response via immune check-point inhibitors for example. The authors should discuss more in detail the challenge of applying "radiomics" in the context of this kind of therapy approach.
A- We agree with the Reviewer, this is really an interesting point, although it is off topic for our manuscript. We have hinted this concept in the conclusions.
Reviewer 3 Report
The authors searched articles about radiology comprehensively and wrote well about radiomics. This paper is important because the correlation between tumors and radiotherapy is described at a glance. but, some careless mistakes were found.
Table.1 in chapter 4.Future directions and conclusions was not accompanied in this manuscript. Prospective trials are rare and important, so it should be included in this paper.
Reference no.189 was not completed. Only authors’ names were written but journal's name, years, and so on were not included.
There was a typo in chapter 3.2. Texture analysis in neoadjuvant radiotherapy - focus on Lung cancer, 6th paragraph. EGRF-inhibitor -> EGFR-inhibitor
Author Response
The authors searched articles about radiology comprehensively and wrote well about radiomics. This paper is important because the correlation between tumors and radiotherapy is described at a glance. but, some careless mistakes were found.
A- We thank the Reviewer for his/her efforts reading end evaluating our manuscript.
Table.1 in chapter 4.Future directions and conclusions was not accompanied in this manuscript. Prospective trials are rare and important, so it should be included in this paper.
- We have added the missing table in the paper.
Reference no.189 was not completed. Only authors’ names were written but journal's name, years, and so on were not included.
- We have corrected the reference.
There was a typo in chapter 3.2. Texture analysis in neoadjuvant radiotherapy - focus on Lung cancer, 6th paragraph. EGRF-inhibitor -> EGFR-inhibitor
- We have corrected the typo.
Round 2
Reviewer 1 Report
Thank you for having revised the manuscript.
Some minor points:
- The title should include radiotherapy otherwise breast, gastric and pancreatic cancer must be included
- English should be improved, particularly in the new sections (some repetitions, spell check, syntahx... for example conversely repeated twice in the new EC paragraph)
- In EC paragraph all the acronyms should be explained (LRLGLe-PET
and RP-CT stand for?). - Minor checks: [18F]FDG vs 18F-FDG vs 18FFDG please conform
- Table 1: add investigators and conform and succint trial design. "Aim of", "purpose of" is implied and could be omitted. I would change "study phase" with "study type" and "disease stage" with "cancer type"
Author Response
Please, see the attachment.
